# MolTextNet: A Two-Million Molecule-Text Dataset for Multimodal Molecular Learning

## Abstract

Small molecules are essential to drug discovery, and graph-language models hold promise for learning molecular properties and functions from text. However, existing molecule-text datasets are limited in scale and informativeness, restricting the training of generalizable multimodal models. We present **MolTextNet**, a dataset of 2.5 million high-quality molecule-text pairs designed to overcome these limitations. To construct it, we propose a synthetic text generation pipeline that integrates structural features, computed properties, bioactivity data, and synthetic complexity. Using GPT-4o-mini, we create structured descriptions for 2.5 million molecules from ChEMBL35, with text over 10 times longer than prior datasets. MolTextNet supports diverse downstream tasks, including property prediction and structure retrieval. Pretraining CLIP-style models with Graph Neural Networks and ModernBERT on MolTextNet yields improved performance, highlighting its potential for advancing foundational multimodal modeling in molecular science.

## 1 Introduction

Small molecules play key roles in scientific discovery for both drug and material development (Edwards et al., 2022; Liu et al., 2024b). A large body of literature describes molecular properties and functions in plain text, motivating the development of machine learning models that jointly understand molecular structures and associated texts Zdrazil et al. (2024). This has driven recent advances in molecule-text multimodal learning (Edwards et al., 2022; Fang et al., 2023; Liu et al., 2024b).

Despite this progress, the development of foundational multimodal molecular models remains limited by the lack of large-scale datasets that pair millions of molecules with diverse and informative descriptions (Fang et al., 2023; Kim et al., 2021; Liu et al., 2024b). Such datasets are essential for enabling generalization across downstream tasks, including property prediction, structure retrieval, and molecule generation from text. Existing molecular textual descriptions are primarily sourced from PubChem, contributed by hundreds of data providers (Kim et al., 2021). However, the number of molecule-text pairs remains limited to about 300K (Fang et al., 2023), with a median description length of only 13 words. For instance, the entry for *1,4-dideoxy-1,4-epithio-D-arabinitol* (structure shown in Figure 1) contains only: *"has been reported in Salacia chinensis with data available,"* which is a description too sparse for models to learn molecular structures or properties. We find that nearly 50% of the dataset consists of similarly uninformative entries.

Informative, large-scale molecule-text datasets should capture three key aspects: structure, properties, and synthesizability, as shown in Figure 1. Each poses a distinct challenge: (1) covering diverse molecular structures across broad chemical spaces for effective pretraining; (2) providing descriptions that reflect structure-property relationships to support tasks like property prediction and inverse design; (3) describing synthetic complexity to enable tasks such as synthetic accessibility estimation, forward and retrosynthetic prediction, and reaction condition inference.

In this work, we propose a synthetic text generation pipeline grounded in computational and experimental molecular annotations. We begin by extracting diverse annotations and summarizing them into coherent molecule-text pairs using GPT-4o-mini (Achiam et al., 2023). Structure-level features are captured via SMARTS-defined functional groups (RDKit Project, 2024). Molecular utility is derived from computed physicochemical properties and over one million bioactivity assays (Zdrazil et al., 2024). To estimate synthetic complexity, we compute heuristic scores and incorporate reaction

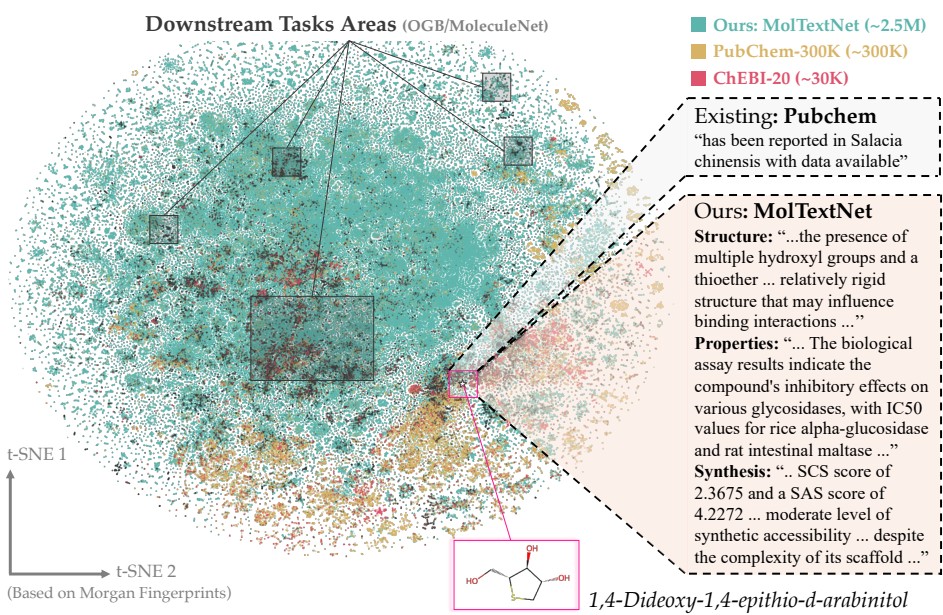

Figure 1: Comparison of PubChem-300K (Fang et al., 2023), ChEBI-20 (Edwards et al., 2021), and MolTextNet. Both PubChem-300K and ChEBI-20 are derived from PubChem (Kim et al., 2021). For reference, we also visualize molecules from commonly used downstream benchmarks (Hu et al., 2020; Wu et al., 2018). Only MolTextNet spans a broader chemical space that covers the structural diversity of these downstream tasks. It also provides more informative descriptions of molecular structures, properties, synthesizability, and their interrelations.

conditions from the USPTO dataset (Coley et al., 2018; Ertl & Schuffenhauer, 2009; Lowe, 2017). Finally, we design a template that integrates all annotations for each molecule, enabling GPT-4o-mini to generate structured scientific descriptions.

By applying our pipeline to the latest ChEMBL release (ChEMBL35, updated on 2024-12-11), we introduce a new dataset, **MolTextNet**. Starting from 2.5 million molecules, 1.7 million assays, and 21 million bioactivities, we generate around 2.5 million molecule-text pairs, as shown in Figures 1 and 2. MolTextNet covers broad chemical space with rich descriptions of molecular structure, properties, and synthesis. On average, the descriptions are over 10 times longer than those in prior datasets, offering a substantial improvement in textual depth. To validate our dataset, we pretrain CLIP-style models using Graph Neural Networks (GNNs) (Xu et al., 2018) and ModernBERT (Warner et al., 2024). Fine-tuning the GNN encoders for property prediction and zero-shot structure retrieval demonstrates the potential of MolTextNet for advancing multimodal molecular learning.

## 2 RELATED WORK

### 2.1 PUBLIC MOLECULE-TEXT DATABASE

Existing textual descriptions of molecules are often sourced from PubChem. Although PubChem contains over 110 million compounds, only a small fraction—approximately 0.28%—have associated textual descriptions, giving rise to datasets such as PCdes (Zeng et al., 2022), PubChemSTM (Liu et al., 2023c), and ChEBI-20 (Degtyarenko et al., 2007; Edwards et al., 2021), many of which contain only brief statements about molecular origin or occurrence. Among these, the version used in Mol-Instructions (Fang et al., 2023) is the largest, comprising approximately 300K molecule-text pairs. We refer to this dataset as PubChem-300K in this work. ChEBI-20 is another subset, focusing on a text-rich part of PubChem that overlaps with the ChEBI database (Degtyarenko et al., 2007).

ChEMBL is another public resource containing manually curated bioactivity data, compiled from over 90K publications. As of version 35 (released on 2024-12-01), it includes 2,496,355 molecules and

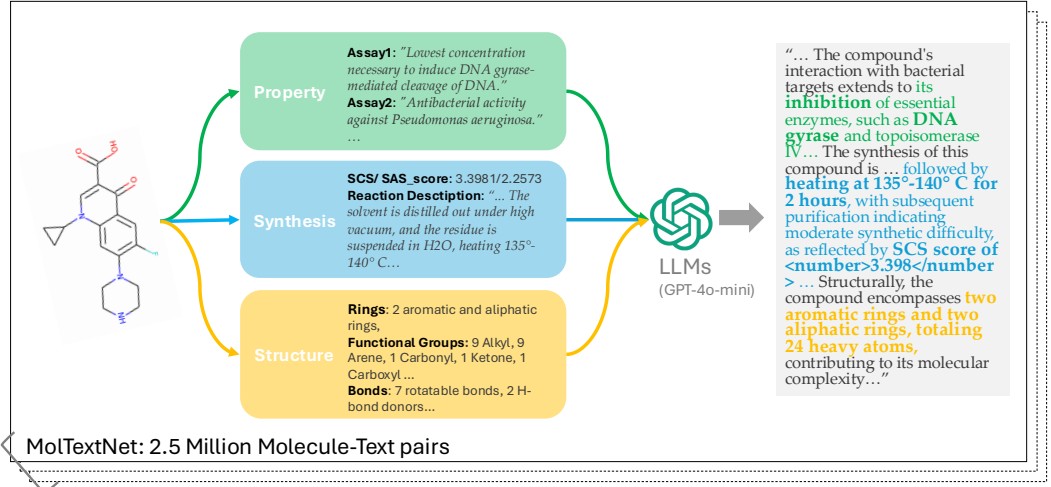

Figure 2: Synthetic Data Generation Pipeline for MolTextNet. Property information is derived from experimental and computational annotations in ChEMBL35 (Zdrazil et al., 2024); synthesis descriptions are generated from heuristic scores and USPTO reaction data (Lowe, 2017). Structural features are extracted using RDKit and approximately 100 predefined functional groups.

approximately 21,123,501 activity records from 1,740,546 assays. While some prior studies (Hu et al., 2019) have used subsets of ChEMBL—such as 456K molecules and 1,410 biochemical assays—for modeling molecule-property relationships, few have utilized the full dataset to capture the complete assay space with textual definitions.

## 2.2 Synthetic Data Generation for Molecules

High-quality pretrained models, such as large language models (LLMs), offer a cost-effective and scalable approach to data generation, and have been widely used to instruct smaller LLMs to follow human prompts (Taori et al., 2023; Wang et al., 2022). Training graph-language multimodal models requires large-scale, aligned molecule-text pairs, which remain underexplored (Liu et al., 2024b). The chemical space is vast, spanning diverse domains across life sciences and materials, yet foundational molecular models for property prediction (Liu et al., 2023a) and structure generation (Liu et al., 2024c) are still lacking. Therefore, we focus on generating synthetic molecular descriptions using LLMs grounded in existing molecular annotations from ChEMBL (Zdrazil et al., 2024), rather than mixing with pseudo-labels as in (Liu et al., 2024b; 2023b).

## 2.3 Multimodal Molecular Learning

Molecular structures can be paired with diverse modalities for multimodal learning, such as 3D protein structures (Schneuing et al., 2024), cellular responses (Liu et al., 2024a), and text descriptions (Edwards et al., 2021; Fang et al., 2023; Liu et al., 2024b; 2023c; Zeng et al., 2022). Among these, text offers a flexible and expressive medium for describing molecules, enabling diverse tasks such as extracting molecular entities from unstructured data (Zeng et al., 2022), captioning molecular structures (Edwards et al., 2022), editing molecules with text prompts (Liu et al., 2023c), and designing molecules guided by textual instructions (Liu et al., 2024b). Existing molecule-text models have shown strong potential and our dataset, MolTextNet, can further unlock their capabilities for building foundational molecular models.

## 3 Methodology of Data Collection

We introduce a synthetic text generation pipeline for molecules, grounded in computational and experimental annotations, and define a prompting template for large language models (LLMs) to rephrase these annotations into scientific descriptions. The overall pipeline is presented in Figure 2.

### 3.1 Preparation of Molecular Annotations

We use all molecules from ChEMBL35 (Zdrazil et al., 2024), each annotated along three dimensions: structure, properties, and synthesizability. The detailed processing procedure is described in Section A.2

**Structure Annotations**   We hypothesize that a compound's biological activity is determined by its chemical scaffold and key functional groups. For each molecule, we extract structures using RDKit, including the Murcko scaffold, ring composition, rotatable bonds, hydrogen bond donors and acceptors, and the presence of over 90 functional groups defined by SMARTS patterns. These features are converted into structured textual phrases in the format "`{count} {structure_name}`," such as "`7 rotatable bonds.`"

**Property Annotations**   We incorporate both computational and experimental annotations. For computational annotations, we extract over 20 physicochemical properties using RDKit (RDKit Project, 2024) and ChemAxon. These include molecular weight, ALogP, polar surface area, rotatable bonds, aromatic ring count, heavy atom count, and drug-likeness scores such as QED and natural product-likeness. Additional descriptors include $pK_a$ values, partition and distribution coefficients, Lipinski rule violations, and compound classification (acidic, basic, or neutral), as recorded in the `COMPOUND_PROPERTIES` table of ChEMBL35. We present the complete table in Table 7.

For experimental annotations, ChEMBL35 has over 1.7 million assays with 21 million associated bioactivity records, covering binding affinity, biological function, ADME, and toxicity. Each assay has a textual definition sourced from the original publication (e.g., "Anticoccidial activity which controlled infection by *Eimeria tenella* in Leghorn cockerels") and standardized activity values with units. We use the `pChEMBL`, i.e., negative logarithm of activity (e.g., $IC_{50}$, $EC_{50}$, $K_i$), and categorize molecules based on thresholds: <5 as "inactive", 5-8 as "slightly active", and >8 as "active".

**Synthesizability Annotations**   We augment each molecule with synthesis-related information by computing two established scores: the Synthetic Complexity Score (SCScore) (Coley et al., 2018), derived from a neural network trained on Reaxys reaction data, and the Synthetic Accessibility Score (SAScore) (Ertl & Schuffenhauer, 2009), which combines fragment contributions and topological complexity. Additionally, we query each molecule against the USPTO reaction dataset (Lowe, 2017). If a match is found, we include the corresponding reaction conditions from the associated patent description.

### 3.2 Synthetic Text Generation with Molecular Annotations and LLMs

We use GPT-4 series models (Achiam et al., 2023) to generate coherent scientific descriptions from molecular annotations. Each molecule is represented as a structured dictionary of property-value pairs, integrating structural features, physicochemical properties, bioactivity profiles, and synthesis information from ChEMBL35 and curated sources. GPT-4o-mini is used for batched generation, while GPT-4o handles samples with high token counts or complex annotations. The template is provided Figure 3.

The models are explicitly prompted to reason over structure-property and structure-synthesis relationships, rather than merely rephrasing or concatenating fields. For example, in Figure 1, the generated description notes the *"presence of multiple hydroxyl groups and a thioether, which enhance solubility in aqueous environments,"* and *"various functional groups such as hydroxyls and thioethers ... which could enhance its biological activity against glycosidases."* illustrating structure-property reasoning. For structure-synthesis relationships, in Figure 2, the model identifies *"two aromatic rings and two aliphatic rings ... contributing to its molecular complexity."* Given the rich structural and property annotations, such relational reasoning enables pretraining of foundational models that map scaffolds, functional groups, and computed descriptors to physicochemical behavior, bioactivity, and synthetic complexity, supporting generalization across diverse downstream tasks.

In addition to prompting the reasoning paths, the model is instructed to provide a formal academic analysis (100-500 words) that strictly describes observed data without summarizing or evaluating; extract relevant factual information concisely. The text must be written as a single plain-text paragraph, avoid repetition, preserve diversity, and exclude unsupported or speculative links. Critical

---

**Prompt Template**

Given a dictionary containing details about a chemical compound, including its name, canonical SMILES string, calculated properties, structural description, biological assay results, and synthetic accessibility, analyze the relationships among structure, properties, complexity, and experimental assay outcomes. \n  {annotation_dictionary} \n
Requirements:

1. Provide a formal academic analysis (100-500 words) that strictly describes observed data without any concluding, summarizing, or evaluative statements.

2. Extract and present the most relevant factual information concisely.

3. Analyze physicochemical behavior, bioactivity, and synthetic complexity by mapping core scaffolds, functional groups, and computed descriptors to molecular interactions, solubility, binding, hydrophobicity, steric effects, and synthetic feasibility, without drawing overall conclusions.

4. Write in plain text as a single paragraph without formatting.

5. Ensure diversity in descriptions and avoid repetition.

6. Keep <number>...</number> format unchanged.

7. State the compound name and canonical SMILES exactly.

8. Ignore missing values and avoid unsupported or speculative links.

9. Exclude introductory phrases such as "Here is the analysis of the polymer...".

---

Figure 3: Prompt template used for generating molecular text grounded in annotations.

tokens—such as SMILES strings, compound names, and numerical values—are preserved exactly as provided, including special <number> tags designed to improve numerical understanding in text. Introductory phrases (e.g., "Here is the analysis...") are excluded, and missing values are ignored.

### 3.3 QUALITY CONTROL

To ensure the quality of synthetic text, we apply specific criteria, filtering rules, and validation steps throughout both the annotation collection and text generation processes.

**Pre-generation**   The original database consists of multiple tables. We extract the canonical SMILES string for each molecule, discard entries with missing or invalid structures (validated using RDKit), and use the ChEMBL identifier `molregno` to deduplicate compounds across tables. Entries with missing values for computed properties or experimental assays are dropped. For fields labeled as "N/A" (i.e., non-null but uninformative), we explicitly instruct the LLM to ignore them. Since ChEMBL provides activity values in various units (e.g., nM, mM), we normalize all concentration-based measurements to nanomolar (nM).

**Long-Text Chunked Processing**   Some entries contain extensive annotations that exceed the 128K-token context window of GPT-4o(-mini). We reserve an 8K-token window for output tokens, resulting in a 120K-token limit for the input tokens, including the system and user prompts. Under this constraint, there are 401 entries that exceed the 120K-token limit, with the maximum length reaching 1.7 million tokens. To feed those entries into LLMs, we chunk the inputs into batches and process them incrementally. The assay dictionary is divided into successive batches that fit within the context limit. For each batch, we prepend the previously generated summary and prompt the model to integrate the new information without modifying or omitting earlier content. This iterative process continues until all assays are incorporated, resulting in a single, coherent summary per molecule.

**Post-generation**   Several rules are applied to validate the output quality after LLM generation. These include checks on description length and consistency between SMILES and compound names. Outputs with insufficient length (e.g., fewer than 100 characters), repetitive patterns, or mismatches

Table 1: Comparison of dataset statistics, including number of pairs, and average/maximum number of words and atoms.

| Dataset | # Molecule-Text Pairs | Words | | Atoms | |
|---|---|---|---|---|---|
| | | Avg. # | Max # | Avg. # | Max # |
| ChEBI-20 | 32,998 | 43.49 | 166 | 32.20 | 574 |
| PubChem-300K | 298,306 | 17.60 | 874 | 33.67 | 574 |
| MolTextNet | 2,474,590 | 253.33 | 1,871 | 30.63 | 780 |

in key fields (e.g., `compound_name`, SMILES) are discarded. Any record failing these checks is regenerated or resubmitted to the API.

## 4 DATASET ANALYSIS

Table 1 summarizes dataset statistics for MolTextNet and existing baselines, while Figure 6 shows joint histograms of molecular size and description length. On average, molecules contain around 30 atoms, but description lengths vary significantly across datasets. Longer descriptions offer greater capacity to convey detailed information. To analyze content diversity, we apply Non-Negative Matrix Factorization (NMF) and Latent Dirichlet Allocation (LDA) to extract latent topics. Topic summaries are shown in Table 2, with full details in Tables 8 and 9. We further group the topics into three categories—structure, property, and synthesizability—and compute the frequency of associated keywords in each molecule-text pair. The normalized values, i.e., the proportions of molecular descriptions containing these keywords, are shown in Figure 4. Details of the categorization are provided in Table 10.

From the tables and figures, ChEBI-20 primarily captures chemical classes such as acid-base species, coenzymes, and fatty acids. While it illustrates structural information well, it falls short in describing properties and synthesizability. PubChem-300K covers a broader range of compounds, including natural products, antibiotics, and synthetic agents, with moderate biological context. Its entries often include synthesis-related information, reflecting molecular availability and supporting synthesizability analysis.

MolTextNet provides the most comprehensive coverage across structural, property, and synthesis dimensions. It contains task-relevant language focused on bioassays, binding affinity, permeability, and molecular property measurements, making it the most suitable dataset for model pretraining.

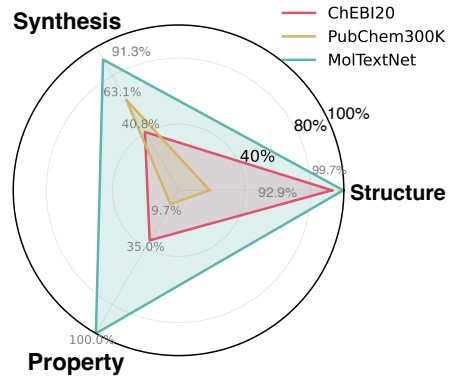

Figure 4: Keyword Coverage (%) in Molecular Descriptions

## 5 DATASET VALIDATION WITH EXPERIMENTS

In this section, we evaluate molecule-text pairs using GNN-BERT-based CLIP models (Radford et al., 2021) to compare MolTextNet against ChEBI-20 and PubChem-300K. We provide both quantitative and qualitative validation of MolTextNet. We randomly sample entries from MolTextNet to match the size of ChEBI-20 and PubChem-300K, constructing two subsets: MolTextNet-50K and MolTextNet-300K, respectively. Dataset statistics are summarized in Tables 1 and 3.

Given molecule-text pairs, we represent molecules as graphs and encode them using a five-layer Graph Isomorphism Network (GIN) (Xu et al., 2018). The GIN is pretrained from scratch. Text descriptions are processed with ModernBERT-Large (Warner et al., 2024), a transformer with an 8192-token context window, well-suited for the long, detailed entries in MolTextNet. The model is pretrained and available on Hugging Face; we continue pretraining its parameters in CLIP models. Its extended capacity allows it to retain long-range dependencies without significant information loss.

Table 2: Topics from LDA and NMF across three molecule-text datasets. Each cell summarizes a topic based on top keywords.

| Topic ID | ChEBI20 | | PubChem300K | | MolTextNet | |
| | LDA | NMF | LDA | NMF | LDA | NMF |
|---|---|---|---|---|---|---|
| 1 | Acid-Base Chemistry | Carboxylic Acid Derivatives | Cancer Cell Inhibitors | Natural Product Metadata | Structure-Activity Relationships | Bioassay Results |
| 2 | Metabolite and Ester Roles | Substituted Agents | Drug Receptor Agents | Antibiotic and Macrocycles | Molecular Targets and Synthesis | Binding and Affinity Evidence |
| 3 | Amino Acids and Derivatives | Coenzyme and Acyl Units | Organic Liquids and Assemblies | Peptides and Linkers | Chemical Fragments and Bioactivity | High-throughput Screen Statistics |
| 4 | Ammonium Inhibitors | Linked Saccharides and Residues | Peptides and Aromatic Compounds | Aromatic and Sugar Assemblies | Antibacterial Activities | Ionization States and pKa Behavior |
| 5 | Fatty Acids and CoA Derivatives | Protonation Chemistry | Microbial Natural Products | Streptomyces-Derived Compounds | Partitioning and Solubility | Partition Coefficients |
| 6 | Acetylated Sugars | Glycerol Derivatives | Microbial Extracts | Functional Fatty Acids | Structure and Binding Profiles | Molecular Weight Estimation |
| 7 | Glycero-phospholipids | Steroidal Positions | Fatty Acid Chemistry | Organic Molecular Classes | Drug-likeness Violations | Cytotoxicity Markers |
| 8 | Drug Agents and Salts | Amino Cations | Steroids and Derivatives | Yeast Metabolites | Binding and Permeability | Antibacterial Sensitivity |
| 9 | Methylated Metabolites | Species-Specific Metabolites | Natural Product Antibiotics | Sulfonamides and Pyridines | Acid-Base Balance | Pathogen Inhibition Assays |
| 10 | Hydroxy-steroids | Fatty Acid Chains | Steroid Functional Groups | Aromatic Substructures | Cellular Assays and Potency | Structural Challenges |

Table 3: Token statistics using ModernBERT and SciBERT tokenizers for CLIP model pretraining.

| Dataset | Tokens (ModernBERT) | | Tokens (SciBERT) | |
| | Avg. # | Max # | Avg. # | Max # |
|---|---|---|---|---|
| ChEBI-20 | 85.33 | 763 | 83.83 | 754 |
| PubChem-300K | 30.27 | 1,308 | 29.46 | 1,278 |
| MolTextNet | 465.00 | 24,603 | 476.72 | 24,576 |
| MolTextNet-50K | 439.62 | 3,162 | 450.40 | 3,214 |
| MolTextNet-300K | 441.82 | 3,162 | 452.73 | 3,214 |

Token limits are set based on the average summary length per dataset: 256 tokens for ChEBI-20 and PubChem-300K, and 1536 tokens for MolTextNet.

We pretrain the GIN-ModernBERT CLIP models for 8 epochs over approximately 2 days on a NVIDIA A6000 GPU. We then evaluate the GIN encoder on downstream property prediction tasks (Section 5.1) and assess both GIN and ModernBERT on zero-shot structure retrieval (Section 5.2). Additionally, we investigate SciBERT as an alternative text encoder in Section 5.3. All pretraining and evaluations are conducted on NVIDIA RTX A6000 GPUs.

Table 4: Fine-tuning performance on seven OGBG classification tasks (Hu et al., 2020): GIN pretrained on MolTextNet-300K consistently achieves the highest AUC($\uparrow$).

| Pretraining Dataset | HIV | ToxCast | Tox21 | BBBP | BACE | ClinTox | SIDER |
|---|---|---|---|---|---|---|---|
| ChEBI-20 | 0.741±0.021 | 0.616±0.015 | 0.732±0.002 | 0.679±0.010 | 0.836±0.011 | 0.885±0.003 | 0.547±0.014 |
| PubChem-300K | 0.752±0.009 | 0.633±0.004 | 0.746±0.002 | 0.686±0.011 | 0.840±0.006 | 0.890±0.010 | 0.602±0.078 |
| MolTextNet-50K | 0.768±0.020 | 0.635±0.002 | 0.744±0.007 | 0.695±0.003 | 0.841±0.000 | 0.886±0.026 | 0.621±0.068 |
| MolTextNet-300K | **0.778±0.010** | **0.638±0.003** | **0.751±0.002** | **0.712±0.004** | **0.847±0.001** | **0.900±0.002** | **0.640±0.031** |

Table 5: Fine-tuning performance on three OGBG regression tasks (Hu et al., 2020): GIN pretrained on MolTextNet-300K consistently achieves the highest $R^2$ and lowest RMSE.

| Pretraining Dataset | MolSol | | MolFreeSol | | MolLipo | |
|---|---|---|---|---|---|---|
| | $R^2 \uparrow$ | RMSE $\downarrow$ | $R^2 \uparrow$ | RMSE $\downarrow$ | $R^2 \uparrow$ | RMSE $\downarrow$ |
| ChEBI-20 | 0.693±0.009 | 1.171±0.017 | 0.543±0.136 | 2.496±0.395 | 0.358±0.169 | 0.876±0.112 |
| PubChem-300K | 0.697±0.008 | 1.164±0.016 | 0.563±0.044 | 2.439±0.150 | 0.474±0.016 | 0.797±0.012 |
| MolTextNet-50K | 0.701±0.033 | 1.161±0.066 | 0.547±0.031 | 2.478±0.105 | 0.503±0.027 | 0.775±0.021 |
| MolTextNet-300K | **0.728±0.016** | **1.106±0.039** | **0.572±0.007** | **2.429±0.019** | **0.531±0.010** | **0.753±0.008** |

## 5.1 DOWNSTREAM TASK 1: MOLECULAR PROPERTY PREDICTION

To validate MolTextNet, we evaluate pretrained GIN encoders on standard molecular property prediction benchmarks from the OGB benchmarks (Hu et al., 2020).vTo avoid data leakage, we removed all overlapping molecules between the OGB benchmarks and the four datasets. The overlap ratios are comparable across datasets of similar sizes (e.g., PubChem-300K and MolTextNet-300K), and in all cases remain below 7%. We use scaffold-based splits to ensure that structurally similar molecules remain within the same split, enabling more rigorous evaluation of generalization.

We use pretrained GIN encoders from ChEBI-20, PubChem-300K, MolTextNet-50K, and MolTextNet-300K, each paired with a lightweight multi-layer perceptron (MLP) prediction head. All models are fine-tuned using the same hyperparameters for 50 epochs with early stopping. We report Area Under the ROC Curve (AUC) for classification tasks and Root Mean Square Error (RMSE) along with the coefficient of determination ($R^2$) for regression. Results are shown in Tables 4 and 5.

We observed that the GIN encoder pretrained on MolTextNet-50K achieves competitive performance across both classification and regression tasks, surpassing ChEBI-20 on all 10 tasks and PubChem-300K on 7 out of 10. Pretraining with more data, as in MolTextNet-300K, further improves the encoder, yielding the best results across all ten tasks after fine-tuning: AUC scores improved by 1-2% on classification tasks, while for the three regression tasks, $R^2$ increased by approximately 6% with corresponding RMSE reductions of 5-10%.

## 5.2 DOWNSTREAM TASK 2: ZERO-SHOT STRUCTURE RETRIEVAL

We validate the zero-shot structure retrieval ability of the pretrained models using test examples from OGBG-MolHIV. Graph representations are generated using pretrained GIN encoders, and structure retrieval queries are formulated as "The molecule has {Number} {Functional Group Name}," then encoded with the text encoders. Molecules are ranked by the similarity between graph and text embeddings. If the number of retrieved functional groups exceeds the required count, accuracy is computed as the ratio of required to retrieved instances. Figure 5 presents the top-1 retrieval results for five queries. Pretrained on MolTextNet-300K, the CLIP models successfully retrieve all queried structures, while ChEBI-20 and PubChem-300K fail in all cases.

## 5.3 ABLATION STUDY ON TEXT ENCODER

Table 6 presents the results of pretraining the CLIP model using SciBERT, a domain-specific encoder optimized for scientific text with a maximum input length of 512 tokens. To accommodate this limitation, text inputs from MolTextNet were truncated to 512 tokens, while all other experimental settings remained constant. Both MolTextNet-50K and MolTextNet-300K outperform ChEBI-20 and PubChem-300K, demonstrating the positive impact of MolTextNet. However, scaling

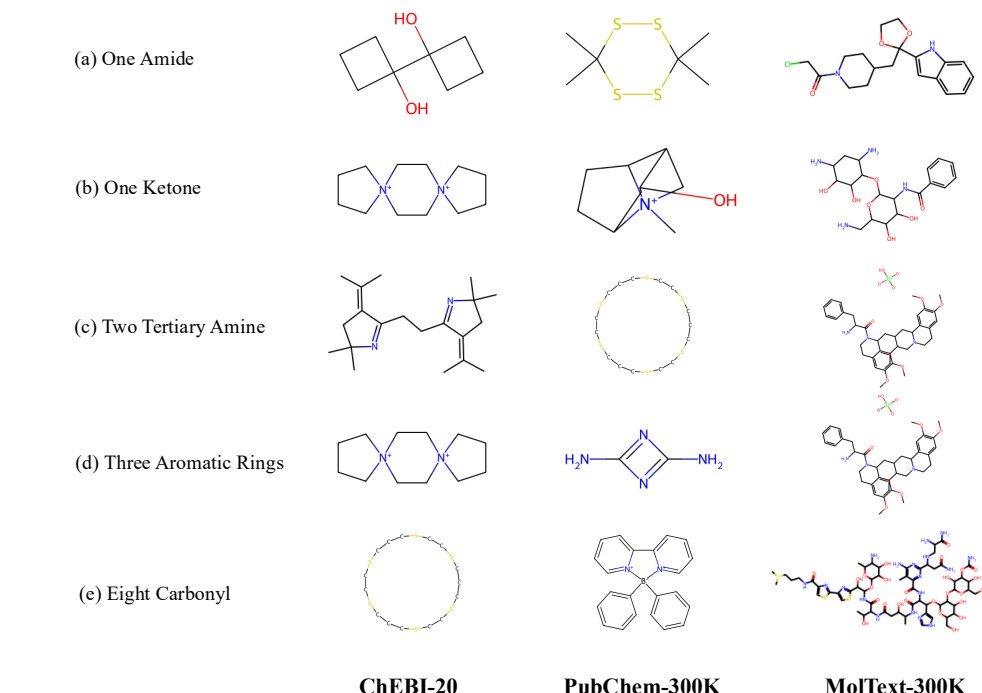

Figure 5: Top-1 structure retrieval results on five functional groups: GIN pretrained on MolTextNet-300K consistently retrieve the right structure described in queries.

Table 6: Fine-tuning performance of the GIN encoder pretrained with the SciBERT text encoder.

| | HIV | Tox21 | BBBP | ClinTox | Molsol | | Mollipo | |
|---|---|---|---|---|---|---|---|---|
| | AUC ↑ | AUC ↑ | AUC ↑ | AUC ↑ | $R^2$ ↑ | RMSE ↓ | $R^2$ ↑ | RMSE ↓ |
| ChEBI-20 | 0.760±0.016 | 0.723±0.007 | 0.674±0.014 | 0.896±0.017 | 0.663±0.029 | 1.228±0.052 | 0.474±0.020 | 0.797±0.015 |
| PubChem-300K | 0.757±0.025 | 0.738±0.002 | 0.694±0.003 | 0.893±0.023 | 0.674±0.023 | 1.207±0.052 | 0.452±0.001 | 0.813±0.001 |
| MolTextNet-50K | 0.757±0.011 | 0.735±0.006 | 0.710±0.011 | 0.889±0.010 | 0.688±0.017 | 1.185±0.034 | 0.490±0.024 | 0.785±0.022 |
| MolTextNet-300K | 0.778±0.008 | 0.743±0.007 | 0.695±0.003 | 0.902±0.007 | 0.703±0.021 | 1.155±0.050 | 0.540±0.019 | 0.747±0.018 |

up to MolTextNet-300K yields limited gains on OGBG-MolHIV, likely due to the severe truncation—reducing input length by two-thirds compared to the 1536-token capacity of ModernBERT-Large. These results highlight the importance of using text encoders with sufficient context length when training on long molecular descriptions.

## 6  CONCLUSION

We presented MolTextNet, a 2.5 million molecule-text dataset to support multimodal molecular learning. Built from the complete ChEMBL35 release, the dataset incorporated 21 million bioactivity records spanning 1.7 million assays. We introduced a synthetic text generation pipeline grounded in diverse molecular annotations, ensuring factual alignment with reference data. The resulting dataset covered broader chemical spaces than existing benchmarks and provided richer descriptions of molecular properties and synthesizability. Experimental results validated its effectiveness in property prediction and structure retrieval, establishing a strong foundation for future molecular models.

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

# A  Technical Appendices and Supplementary Material

## A.1  More Details on Molecular Annotations

Table 7: Computed molecular descriptors from ChEMBL based on RDKit and ChemAxon software.

| Calculated Properties | Type | Description |
| --- | --- | --- |
| MW_FREEBASE | Number | Molecular weight of parent compound |
| ALOGP | Number | Calculated ALogP |
| HBA | Number | Number of hydrogen bond acceptors |
| HBD | Number | Number of hydrogen bond donors |
| PSA | Number | Polar surface area |
| RTB | Number | Number of rotatable bonds |
| RO3_PASS | String | Indicates whether the compound passes the rule-of-three (MW < 300, logP < 3, etc.) |
| NUM_RO5_VIOLATIONS | Number | Number of violations of Lipinski's rule-of-five, using HBA and HBD definitions |
| CX_MOST_APKA | Number | The most acidic pKa calculated using ChemAxon v17.29.0 |
| CX_MOST_BPKA | Number | The most basic pKa calculated using ChemAxon v17.29.0 |
| CX_LOGP | Number | The calculated octanol/water partition coefficient using ChemAxon v17.29.0 |
| CX_LOGD | Number | The calculated octanol/water distribution coefficient at pH 7.4 using ChemAxon v17.29.0 |
| MOLECULAR_SPECIES | String | Indicates whether the compound is an acid, base, or neutral |
| FULL_MWT | Number | Molecular weight of the full compound including any salts |
| AROMATIC_RINGS | Number | Number of aromatic rings |
| HEAVY_ATOMS | Number | Number of heavy (non-hydrogen) atoms |
| QED_WEIGHTED | Number | Weighted quantitative estimate of drug-likeness (Bickerton et al., Nature Chem 2012) |
| MW_MONOISOTOPIC | Number | Monoisotopic parent molecular weight |
| FULL_MOLFORMULA | String | Molecular formula for the full compound (including any salt) |
| HBA_LIPINSKI | Number | Number of hydrogen bond acceptors by Lipinski's original rules (N + O count) |
| HBD_LIPINSKI | Number | Number of hydrogen bond donors by Lipinski's original rules (NH + OH count) |
| NUM_LIPINSKI_RO5_VIOLATIONS | Number | Number of violations of Lipinski's rule-of-five using HBA_LIPINSKI and HBD_LIPINSKI |
| NP_LIKENESS_SCORE | Number | Natural product-likeness score (Ertl et al., J. Chem. Inf. Model., 2008) |

The full list of computable properties is shown in Table 7. These properties are also available in the ChEMBL35 database.

The functional groups considered include Alkyl, Alkene, Alkyne, Arene, Carbonyl, Aldehyde, Ketone, Carboxyl, Ester, Amide, Anhydride, Acyl Halide, Hydroxyl, Phenol, Enol, Ether, Thiol, Sulfoxide, Sulfone, Sulfonic Acid, Sulfonamide, Nitrile, Nitro, Azide, Diazo, Azo, Hydrazone, Oxime, Imine, Azomethine, Hydroxylamine, Hydrazine, Hydrazide, Iminium, Carbamate, Cyanamide, N-Oxide, Peroxide, Phosphate, Sulfate, Primary Amine, Secondary Amine, Tertiary Amine, Thioether, Disulfide, Thioester, Sulfinic Acid, Sulfonate Ester, Sulfamate, Sulfamide, Isocyanate, Isothiocyanate, Urea, Guanidine, Carbodiimide, Phosphine, Phosphonic Acid, Phosphonate Ester, Phosphoramidate, Phosphoramide, Phosphonamide, Phosphine Oxide, Phosphite, Phosphonite, Phosphoramidite, Phosphoramidate, Phosphinate, Boronic Acid, Boronate Ester, Boronic Ester, Silyl Ether, Silanol, Silyl Halide, Alkyl Halide, Aryl Halide, Perfluoroalkyl, Epoxide, Lactone, Lactam, Semicarbazide, Aziridine, Azepane, Aminal, Thioamide, Sulfenic Acid, Sulfinyl, and Sulfonyl.

## A.2 ChEMBL Processing Procedure

We construct MolTextNet starting from ChEMBL35, a database maintained by the European Bioinformatics Institute (EMBL-EBI) that integrates chemical structures, biological activity data, and genomic information. The latest release contains approximately 2.4 million distinct small molecules, 20.8 million bioactivity measurements, and over 1.6 million assays. Below, we describe our pipeline for constructing a molecule-text dataset using curated molecular annotations and high-quality generated descriptions.

### A.2.1 Database Filtering

ChEMBL35 is distributed in various formats—including MySQL, PostgreSQL, SQLite dumps; SDF structure files; FASTA sequences; and RDF triples—each exposing a molecule → structure → activity → assay relational schema. We use the MySQL release, which includes 65 tables and over 100 million rows, to extract high-quality molecular samples.

**SMILES Validation**  Canonical SMILES strings are used as the molecular graph input for downstream GNNs. We extract each molecule's SMILES and `compound_name`, discard missing or RDKit-invalid entries, and collapse duplicates using the ChEMBL identifier `molregno` to ensure one representative entry per molecule.

**Information Curation**  For each validated molecule, we extract compound-level physicochemical properties—such as molecular weight, ALogP, HBA/HBD counts, PSA, rotatable bonds, Rule-of-Three/Five compliance, $pK_a$/$pK_b$, and QED—from the `compound_properties` table. These are joined with other tables (e.g., `activities`, `assays`) to collect quantitative assay endpoints with normalized units. Qualitative or unit-less values are excluded, and missing data is dropped. Because one molecule may be associated with multiple assays, we group all assay-level descriptions and measurements under the parent molecule, preserving full experimental context.

This yields approximately 2.4 million JSON-encoded entries, each containing a sanitized SMILES string, compound name, physicochemical properties, and assay metadata with experimental results and descriptions.

### A.2.2 Dataset Post-processing

After constructing the initial dataset, we apply post-processing steps to enrich each JSON entry with standardized annotations, structural summaries, and synthesis metrics.

**Additional Information**

- **Bioactivity:** For each assay, we extract the human-readable `action_type` and map the associated pChEMBL value into three categories: "not active" (pChEMBL $< 5$), "slightly active" ($5 \leq$ pChEMBL $< 8$), and "active" (pChEMBL $\geq 8$). This provides a unified scale for biological activity.

- **Structure:** We incorporate structured summaries to reflect the hypothesis that biological activity is influenced by a molecule's scaffold and functional groups. For each SMILES, we extract the Bemis-Murcko scaffold, ring counts, H-bond donors/acceptors, rotatable bonds, and functional group frequencies (using SMARTS patterns), and convert these into descriptive sentences.

- **Synthesis:** We compute synthesis-related metrics, including the Synthetic Complexity Score (SCScore), obtained from a neural network trained on Reaxys reactions (Coley et al., 2018), and the Synthetic Accessibility Score (SAScore) (Ertl & Schuffenhauer, 2009), which combines fragment contributions with topological features. Additionally, we match molecules to USPTO reaction precedents to include synthesis conditions where available.

**Numeric Tagging**  To preserve quantitative content during generation, all numeric fields (e.g., bioactivity values) are wrapped in `<number>...</number>` markers, enabling the model to distinguish numerical values from surrounding text.

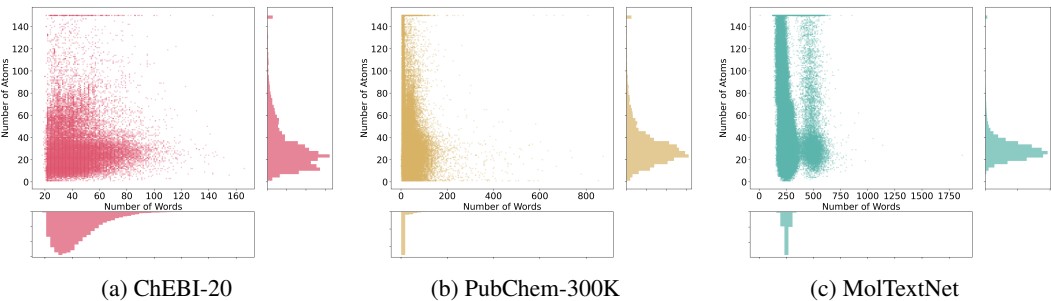

Figure 6: Joint histograms of word and atom counts for different datasets.

### A.3 MORE DETAILS ON DATASET ANALYSIS

Figure 6 shows joint histograms of word and atom counts for MolTextNet, ChEBI-20, and PubChem-300K. Most descriptions in ChEBI-20 contain fewer than 100 words, and those in PubChem-300K fewer than 200. In contrast, MolTextNet predominantly contains descriptions ranging from 250 to 500 words, indicating that the LLMs effectively follow length-specific generation instructions.

### A.4 MORE DETAILS ON EXPERIMENTAL SETUPS

Given the substantial size of the MolTextNet dataset, we adopt a memory-efficient data loading strategy. The full corpus is preprocessed and stored in HDF5 format, partitioned into several shards of 50K samples each. During training, we implement an on-demand loading mechanism that dynamically reads only the relevant shard into memory for the current epoch. This design ensures full dataset coverage across epochs while effectively mitigating out-of-memory issues, thereby enabling large-scale training on resource-constrained environments.

For downstream tasks, we adopt the standard molecular property prediction benchmarks from the OGB dataset Hu et al. (2020), following the original scaffold-based train/validation/test split for consistent evaluation. Molecular property prediction is conducted by fine-tuning pretrained GIN encoders with a 2-layer MLP for 50 epochs, using early stopping with a patience of 10 epochs. The MLP learning rate is fixed to 1e-3, while the GIN encoder learning rate is set as 1e-3 or 1e-4, with a drop ratio of 0 or 0.1. To ensure fidelity, all pretrained models share a unified hyperparameter configuration across tasks. For the zero-shot structure retrieval task, the pretrained GIN encoders directly encode SMILES strings, which are then matched against the embeddings of the query text generated by the pretrained text encoders. Detailed query texts and SMILES mappings are provided in Section A.6.

### A.5 MORE DETAILS ON TOPIC MODELING OF MOLECULAR DESCRIPTIONS

To evaluate which dataset is most suitable for pretraining molecular language models, we analyzed the topic keywords extracted from ChEBI-20, PubChem-300K, and MolTextNet using both LDA and NMF. The full topic lists are presented in Tables 8 and 9. We further group these keywords into three categories, as shown in Table 10, to highlight the different dimensions present in molecular descriptions.

From the tables, ChEBI-20 predominantly contains ontology-style terms related to basic chemical groups (e.g., `acid`, `anion`, `carboxylic`) and shows limited lexical variation and minimal coverage of molecular effects. PubChem-300K offers greater diversity, including references to both biosourced and synthetic molecules (e.g., `streptomyces`, `macrolide`, `antibiotic`), with moderate coverage of experimental conditions.

In contrast, MolTextNet exhibits the richest and most varied language, with terms describing assay protocols, molecular properties, and activity patterns (e.g., `assays`, `partition`, `inhibition`, `affinity`, `suggesting`), as well as detailed experimental contexts (e.g., `MIC`, $\text{IC}_{50}$, `cytotoxicity`, `partition coefficient`, `synthetic route`). It also includes structure-aware terms (e.g., `likeness`, `violations`, `ccc`, `structural`) that are likely bene-

Table 8: Keywords and topic proportions from LDA on three molecular text datasets.

| Topic | ChEBI-20 | | PubChem-300K | | MolTextNet | |
|---|---|---|---|---|---|---|
| | Keywords | Prop. | Keywords | Prop. | Keywords | Prop. |
| 1 | conjugate, base, acid, anion, major, pH, deprotonation, species, obtained, group | 13.4% | cell, activity, inhibitor, cells, tumor, compound, antineoplastic, inhibits, produced, kinase | 5.2% | cc, suggesting, properties, level, influence, structural, activity, inhibition, binding, targets | 9.3% |
| 2 | metabolite, acid, role, derives, human, group, hydroxy, ester, formal, condensation | 10.0% | used, treatment, drug, agent, receptor, inhibitor, polysaccharide, antagonist, activity, effects | 5.2% | cc, activity, binding, multiple, suggests, nm, targets, complex, synthesis, ccc | 15.3% |
| 3 | acid, amino, conjugate, alpha, group, monocarboxylic, derives, derivative, hydroxy, tautomer | 10.7% | compound, sn, used, water, organic, glycero, ring, liquid, assembly, chemical | 5.5% | cc, nc, nm, yl, ccc, ic, human, methyl, activity, amino | 8.1% |
| 4 | amino, group, cation, role, organic, ion, acid, derivative, ammonium, inhibitor | 6.6% | member, peptide, aromatic, ether, benzenes, oligopeptide, amide, biphenyls, amine, tripterygium | 6.7% | ml, cc, activity, µg, mic, strains, antibacterial, inhibitory, suggesting, exhibits | 3.5% |
| 5 | coa, fatty, acid, acyl, chain, group, long, conjugate, trans, hydroxy | 6.3% | product, natural, available, data, streptomyces, aspergillus, organisms, carbohydrate, derivatives, carbohydrates | 13.1% | coefficient, cc, suggesting, water, octanol, properties, targets, partition, inhibition, structural | 8.9% |
| 6 | beta, alpha, acetyl, amino, residue, consisting, residues, glucosamine, oligosaccharide, linked | 9.6% | product, natural, available, data, organisms, penicillium, japonica, artemisia, isodon, indica | 31.9% | nm, assays, cc, sid, targets, suggesting, activity, influence, properties, structural | 14.0% |
| 7 | acyl, sn, acid, phosphate, glycero, derives, specified, groups, glycerol, respectively | 5.8% | acid, conjugate, base, fatty, group, metabolite, lactam, azamacrocycle, acyl, related | 10.4% | likeness, drug, quantitative, estimate, weighted, suggesting, violations, structural, absence, activity | 4.9% |
| 8 | agent, role, inhibitor, salt, drug, used, contains, anti, ec, antagonist | 9.5% | member, steroid, glycoside, acids, salt, role, contains, ureas, ester, hydroxy | 7.0% | targets, binding, properties, suggesting, favorable, suggests, activity, enhance, permeability, structural | 11.3% |
| 9 | member, group, position, compound, role, substituted, methyl, class, metabolite, positions | 16.6% | natural, product, available, data, sulfonamide, euphorbia, triglyceride, organisms, piper, laurencia | 5.6% | cc, pka, ccc, suggesting, basic, nc, influence, acidic, value, nm | 15.8% |
| 10 | hydroxy, metabolite, role, beta, steroid, position, isolated, derives, group, alpha | 11.4% | role, beta, alpha, metabolite, group, position, amino, compound, related, functionally | 9.4% | cc, nm, cells, activity, ic, oc, human, suggesting, exhibits, assays | 9.1% |

Table 9: Keywords and normalized topic proportions from NMF on three molecular text datasets.

| Topic | ChEBI-20 | | PubChem-300K | | MolTextNet | |
|---|---|---|---|---|---|---|
| | Keywords | Prop. | Keywords | Prop. | Keywords | Prop. |
| 1 | acid, monocarboxylic, conjugate, derives, group, carboxy, dicarboxylic, carboxylic, amino, formal | 10.95 | data, product, natural, available, organisms, aspergillus, penicillium, euphorbia, artemisia, japonica | 25.94 | sid, nm, inconclusive, assays, potency, named, results, representation, inactive, inhibitors | 9.82 |
| 2 | member, position, group, substituted, compound, methyl, agent, class, positions, inhibitor | 12.38 | azamacrocycle, lactam, sulfate, macrolide, role, beta, gamma, antibiotic, metabolite, agent | 4.28 | receptor, activity, binding, suggests, multiple, enhance, likely, affinity, potentially, indicates | 18.90 |
| 3 | coa, acyl, coenzyme, diphosphate, thiol, results, condensation, formal, phosphate, fatty | 6.25 | peptide, cyclic, role, composed, joined, metabolite, linkages, sequence, leucine, tripeptide | 3.95 | mmv, percentage, nf, nanoglo, μm, hours, primary, unknown, screen, remains | 9.63 |
| 4 | beta, alpha, acetyl, amino, residue, glucosamine, oligosaccharide, trisaccharide, consisting, linked | 10.37 | member, ureas, benzenes, assembly, ring, quinolines, carbohydrates, biphenyls, derivatives, carbohydrate | 7.64 | pka, basic, acidic, physiological, conditions, ionization, state, suggesting, states, protonation | 14.72 |
| 5 | base, conjugate, anion, deprotonation, pH, major, species, obtained, carboxy, phosphate | 10.80 | streptomyces, data, product, natural, available, albidoflavus, hygroscopicus, griseus, platensis, albus | 4.09 | coefficient, water, octanol, partition, distribution, pH, hydrophobic, supported, parent, atoms | 8.76 |
| 6 | sn, acyl, glycero, phosphate, specified, glycerol, oleoyl, diacyl, groups, respectively | 6.37 | acid, amino, conjugate, fatty, group, base, functionally, related, hydroxy, chain | 7.95 | likeness, drug, estimate, weighted, quantitative, absence, supports, atoms, heavy, violations | 9.95 |
| 7 | steroid, hydroxy, beta, oxo, alpha, delta, hydride, derives, position, positions | 6.66 | compound, glycosyl, carbonyl, organooxygen, organonitrogen, organic, amino, organohalogen, functionally, related | 3.85 | nm, cells, ic, human, oc, cell, values, lines, cytotoxicity, yl | 12.05 |
| 8 | cation, organic, amino, ion, ammonium, protonation, derivative, conjugate, obtained, tertiary | 7.02 | metabolite, produced, saccharomyces, cerevisiae, escherichia, coli, strain, mg, role, human | 4.19 | ml, μg, mic, antibacterial, minimum, strains, staphylococcus, inhibitory, aureus, ug | 5.37 |
| 9 | metabolite, role, human, mouse, plant, cerevisiae, saccharomyces, coli, escherichia, derives | 13.61 | sulfonamide, benzenes, antibiotic, group, role, used, antibacterial, agent, inhibitor, pyridines | 2.06 | ddd, inhibition, percentages, stage, falciparum, um, hepg, leishmania, targets, assays | 8.73 |
| 10 | fatty, chain, long, acid, hydroxy, anion, omega, polyunsaturated, saturated, branched | 5.69 | aromatic, ether, amide, ketone, amine, flavonoids, benzenoid, amino, furans, thiophenes | 3.05 | nc, cc, ccc, yl, challenges, ccccc, amino, significant, oral, high | 13.38 |

Table 10: Keyword sets for each semantic dimension (structure, property or synthesizability) used in description categorization.

| Dimension | Structure | Property | Synthesizability |
|---|---|---|---|
| Keywords | conjugate, base, acid, anion, ph, deprotonation, species, group, amino, alpha, beta, monocarboxylic, derivative, hydroxy, tautomer, cation, organic, ion, ammonium, acyl, phosphate, glycero, glycerol, sn, position, substituted, methyl, class, steroid, ring, liquid, assembly, yl, nc, ccc, pka, value, basic, acidic, coefficient, octanol, partition, structural | cell, activity, inhibitor, tumor, compound, antineoplastic, inhibits, kinase, receptor, drug, treatment, agent, antagonist, effects, binding, suggests, suggesting, targets, multiple, µg, mic, strains, antibacterial, inhibitory, exhibits, assays, nm, ic, oc, human, likeness, quantitative, estimate, weighted, violations, enhance, permeability, favorable, cells | coa, fatty, acyl, chain, long, trans, residue, residues, acetyl, glucosamine, oligosaccharide, linked, product, natural, available, data, streptomyces, aspergillus, penicillium, organisms, carbohydrate, carbohydrates, japonica, artemisia, isodon, indica, biosynthetic, contains, salt, ureas, glycoside, ec, related, complex, synthesis |

ficial for generative modeling. These findings suggest that MolTextNet provides the most comprehensive linguistic and contextual grounding for pretraining models across diverse downstream tasks, including property prediction, structure generation, and reaction condition inference.

A.6   MORE RESULTS ON ZERO-SHOT STRUCTURE RETRIEVAL

We defined 7 case studies to retrieve multiple functional groups. Their query texts are defined as:

- **Case 1**: The molecule has one Amide group,
- **Case 2**: The molecule has one Ketone group,
- **Case 3**: The molecule has one Primary Amine group,
- **Case 4**: The molecule has two Tertiary Amine groups,
- **Case 5**: The molecule has three Aromatic Rings,
- **Case 6**: The molecule has four Ester groups,
- **Case 7**: The molecule has eight Carbonyl groups,

Functional group-SMILES mapping is:

- Amide: [NX3][CX3](=O)[#6],
- Ketone: [CX3](=O)[#6],
- Primary Amine: [NX3H2],
- Tertiary Amine: [NX3]([#6])([#6])[#6],
- Aromatic Ring: [c],
- Ester: [CX3](=O)[OX2H0][#6],
- Carbonyl: [CX3]=O.

For ChEBI-20, PubChem-300K, MolTextNet-300K, their top-3 retrieved results are visualized in Figures 7 to 13.

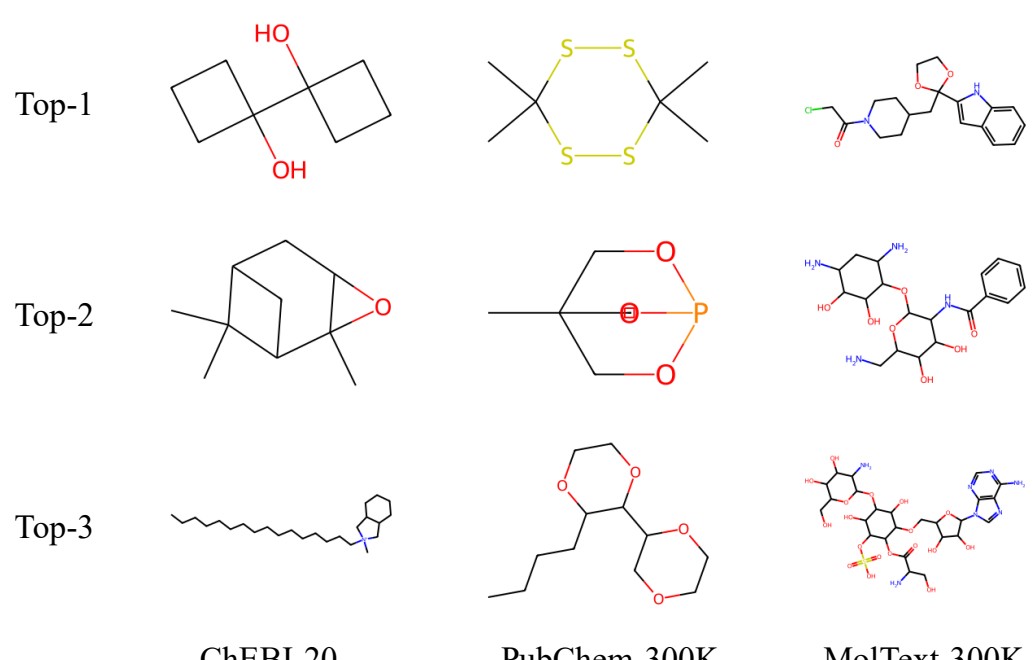

|  | ChEBI-20 | PubChem-300K | MolText-300K |

Figure 7: Top-3 structure retrieval results on Case 1 (The molecule has one Amide group): GIN pretrained on MolTextNet-300K consistently retrieve the right structure described in the query.

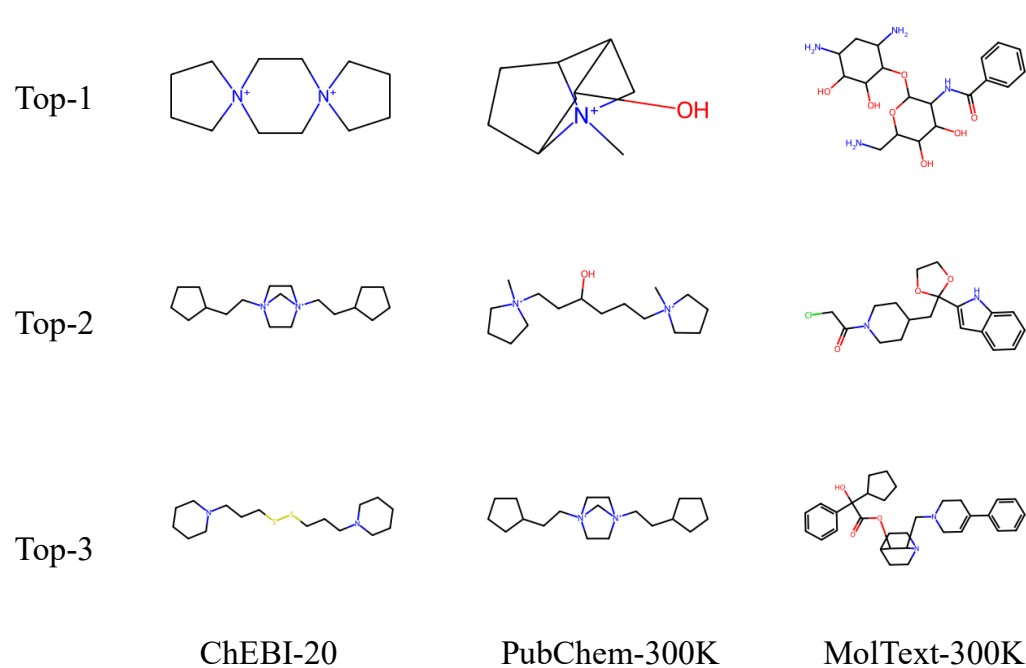

Figure 8: Top-3 structure retrieval results on Case 2 (The molecule has one Ketone group): GIN pretrained on MolTextNet-300K consistently retrieve the right structure described in the query.

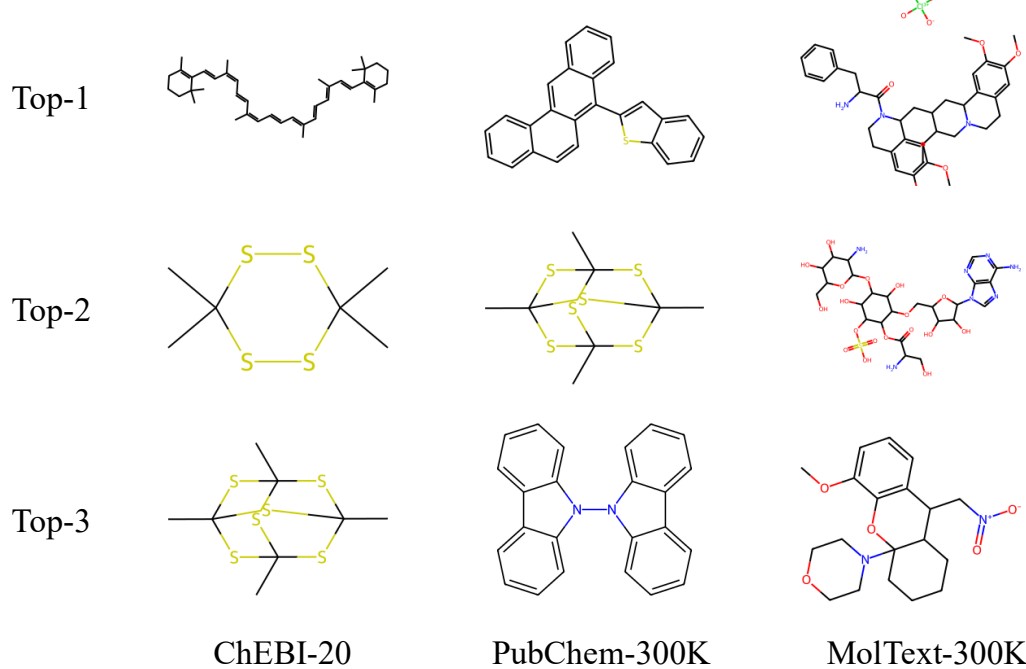

Figure 9: Top-3 structure retrieval results on Case 3 (The molecule has one Primary Amine group): GIN pretrained on MolTextNet-300K consistently retrieve the right structure described in the query.

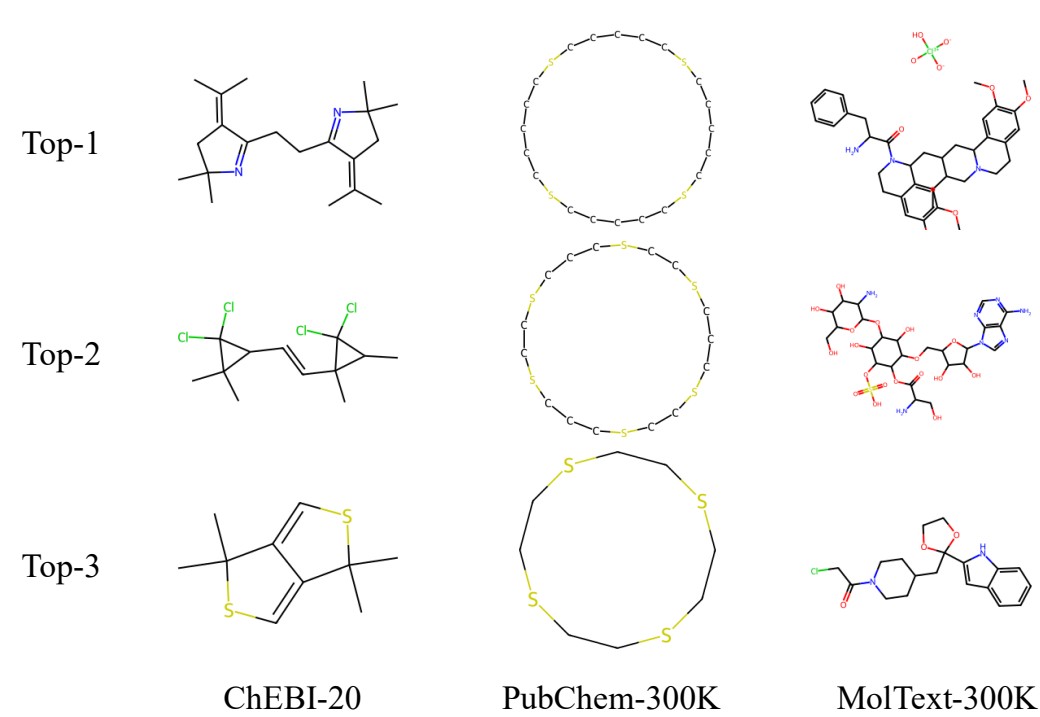

Figure 10: Top-3 structure retrieval results on Case 4 (The molecule has two Tertiary Amine groups): GIN pretrained on MolTextNet-300K consistently retrieve the right structure described in the query.

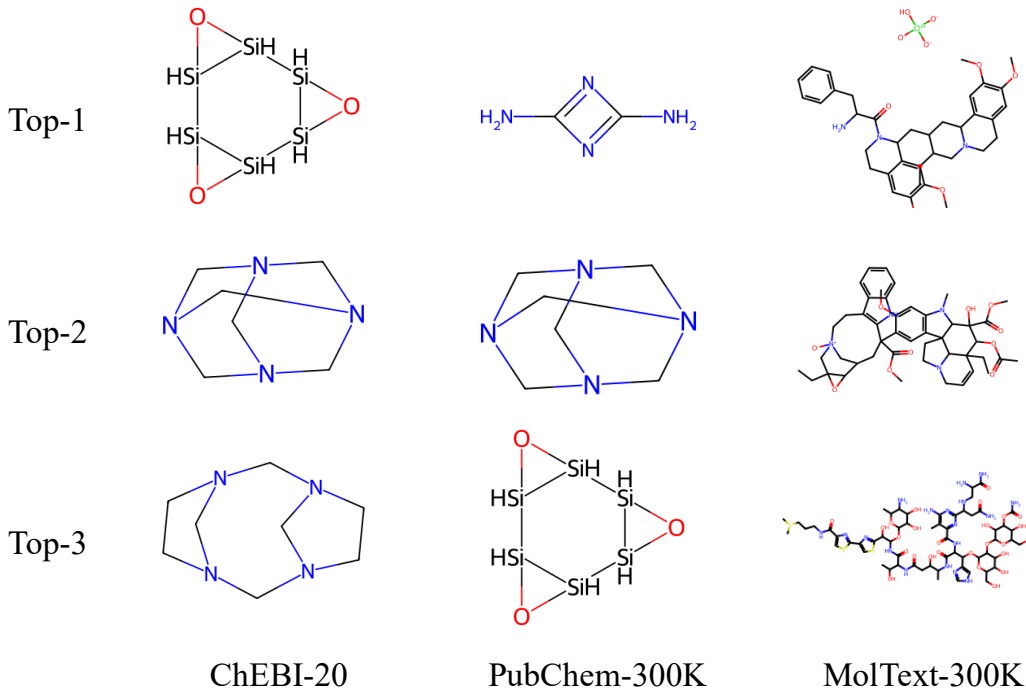

Figure 11: Top-3 structure retrieval results on Case 5 (The molecule has three Aromatic Rings): GIN pretrained on MolTextNet-300K consistently retrieve the right structure described in the query.

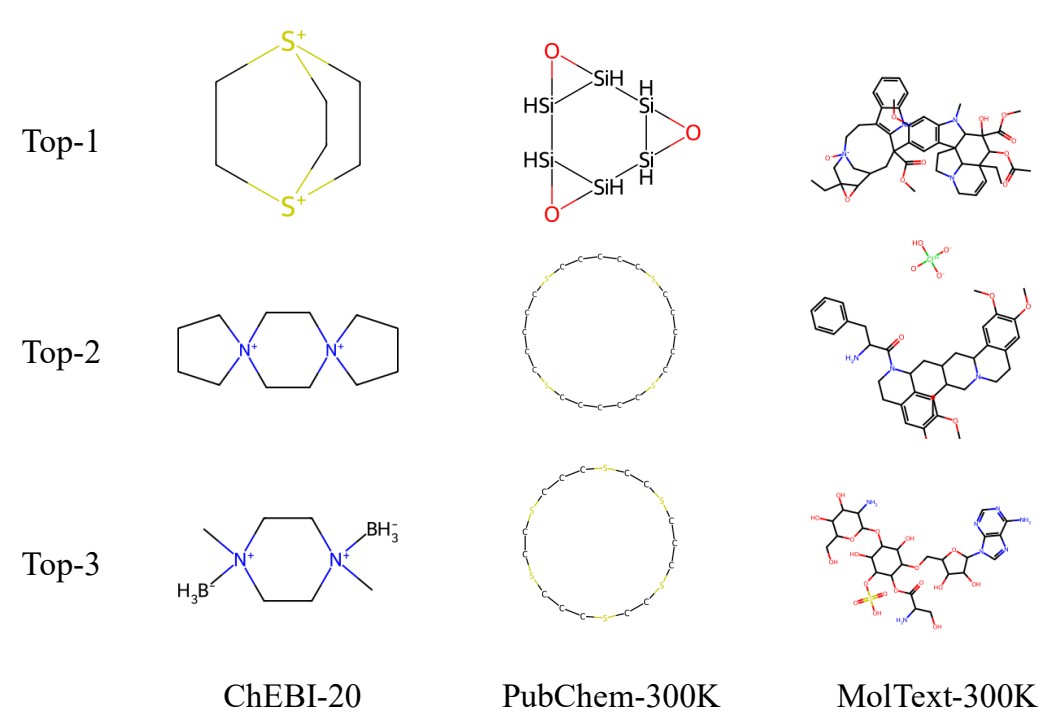

Figure 12: Top-3 structure retrieval results on Case 6 (The molecule has four Ester groups): GIN pretrained on MolTextNet-300K consistently retrieve the right structure described in the query.

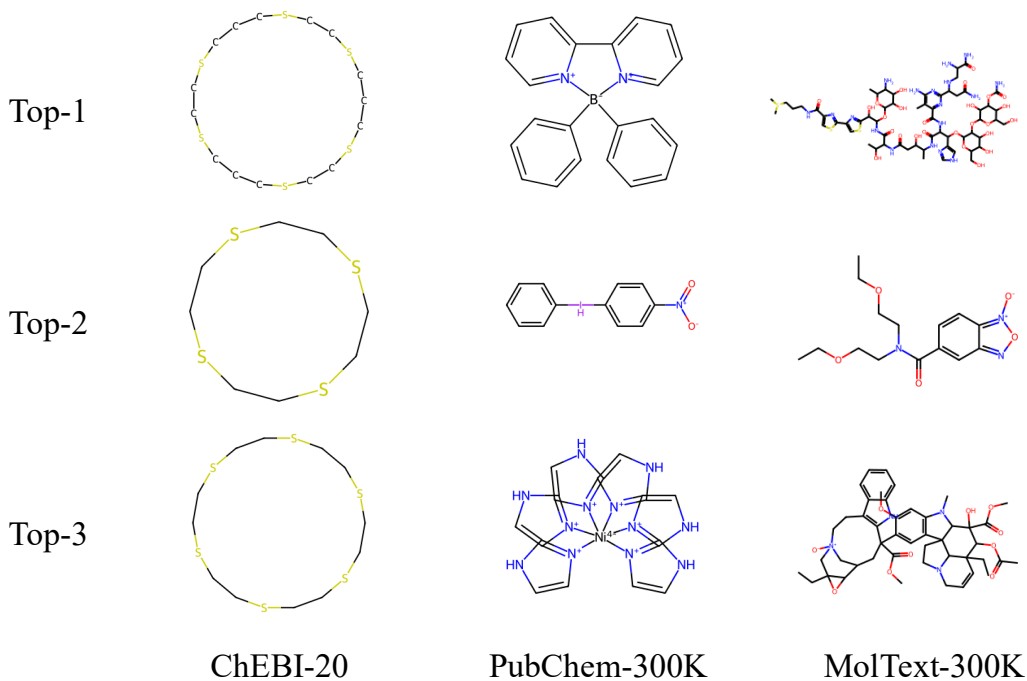

Figure 13: Top-3 structure retrieval results on Case 7 (The molecule has eight Carbonyl groups): GIN pretrained on MolTextNet-300K consistently retrieve the right structure described in the query.

