# OpenReview forum: "MolTextNet: A Two-Million Molecule-Text Dataset for Multimodal Molecular Learning"
_ICLR.cc/2026/Conference — ICLR 2026 Conference Withdrawn Submission_

### Official Review · Reviewer_NFq3 · 2025-10-29

**Soundness:** 3
**Presentation:** 3
**Contribution:** 3
**Rating:** 6
**Confidence:** 3

**Summary:**

The paper introduces MolTextNet, a large-scale molecule–text dataset built from ChEMBL35 and USPTO-derived cues. The authors generate approximately 2.5M molecule–text pairs by prompting GPT-4o-mini to synthesize structured, 100–500-word scientific descriptions that tie structural motifs, physicochemical properties, assay outcomes, and synthesis difficulty together; critical tokens (names, SMILES, numbers) are preserved and unit normalization plus filtering are applied to control quality. Compared with PubChem-300K and ChEBI-20, MolTextNet offers longer, denser text and broader coverage. The paper validates the dataset by pretraining CLIP-style models using a GIN graph encoder and a long-context ModernBERT text encoder, then fine-tuning on OGB classification and regression tasks and testing zero-shot structure retrieval formed as functional-group queries. Across 10 downstream tasks the models pretrained on MolTextNet-300K subsets generally outperform those trained on prior datasets, with additional analyses on text encoders and topic coverage.

**Strengths:**

- MolTextNet reports about 2.5M pairs with far longer descriptions than PubChem-300K and ChEBI-20, with explicit attempts to include structure, property, and synthesis dimensions. This improves lexical and conceptual grounding for multimodal pretraining.

- The paper details how SMILES and compound names are validated, how properties and assays are collected and normalized, how numeric tokens are preserved, and how chunking is handled for ultra-long entries. This level of detail supports reuse.

- Consistent gains when pretraining on MolTextNet subsets are shown across seven OGB classification and three regression tasks, and the functional-group retrieval task highlights alignment between text and structure learned by CLIP-style training.

- Using ModernBERT for long context shows why a longer-context encoder matters; the SciBERT ablation underscores the cost of truncation. Compute and training settings are stated.

**Weaknesses:**

- The paper relies on post-generation rules (length, token consistency) but does not quantify error rates for numeric values, assay units, or misattributed functional groups. A small-scale human or programmatic audit (e.g., SMILES-parsed substructure counts vs. described counts, unit conversions) would strengthen trust in the synthetic text. (major)

- The zero-shot retrieval task is built from simple functional-group templates; it does not test richer assay semantics, property trends, or synthesis narratives. Broader retrieval formulations (e.g., “two para-substituted phenyl rings and high polar surface area”) or caption-to-graph matching on held-out families could better reflect the dataset’s intent.

- Only a GIN encoder is studied on the graph side; modern graph transformers or message-passing variants could show whether the benefit generalizes across inductive biases. The text side focuses on two encoders; more comparisons (e.g., Longformer-style scientific encoders) would provide a fuller picture.

- Coverage claims would be stronger with quantitative chemical-space measures beyond t-SNE and topic models, such as Bemis–Murcko scaffold diversity, ECFP-based internal diversity, pairwise Tanimoto distributions against downstream tasks, and assay-type entropy. Current plots are informative but not sufficient to support generalization claims on their own.

**Questions:**

1. Can you report a measured error rate for numeric fields and substructure mentions by cross-checking generated text against RDKit-computed counts and standardized pChEMBL values after unit conversion? A 1–2% sampled audit with confidence intervals would help.

1. You state overlap with OGB was removed and remains below 7%. Please provide the exact overlap counts per OGB dataset, the hashing or SMILES canonicalization used, and an independent hash list to enable external verification.

1. For the 401 ultra-long entries, how often did iterative summarization change earlier facts? Do you have a checksum-based or programmatic equivalence test to ensure no loss or drift across chunks?

1. Can you add queries that combine structural patterns with property statements (e.g., “three aromatic rings and high ALogP with low HBD”) and report mAP/Recall@k to demonstrate semantic alignment beyond pure substructure counting?

1. Do the MolTextNet gains transfer to Graphormer or PSP-style graph transformers? Even a small subset experiment would help position the dataset for broader use.

---

### Official Review · Reviewer_uLf8 · 2025-10-30

**Soundness:** 3
**Presentation:** 3
**Contribution:** 3
**Rating:** 4
**Confidence:** 5

**Summary:**

This paper introduces MolTextNet, a large-scale dataset consisting of approximately 2.5 million molecule–text pairs constructed from ChEMBL35. The authors develop a synthetic text generation pipeline using GPT-4o-mini to produce long, structured descriptions that integrate molecular structure, physicochemical properties, bioactivity assays, and synthesis complexity. Compared with existing resources such as PubChem-300K and ChEBI-20, MolTextNet provides significantly longer and richer textual annotations, improving coverage across chemical space. The authors validate the dataset by pretraining CLIP-style models combining GIN and ModernBERT, showing consistent gains on standard property-prediction benchmarks (OGB tasks) and demonstrating better zero-shot structure retrieval. The work highlights the potential of synthetic molecule-text data for multimodal molecular modeling and aims to provide a new foundation for future research.

**Strengths:**

1.Presents the largest molecule–text dataset so far, combining structure, property, and synthesis aspects.
2.The data pipeline is transparent and well-documented, facilitating reproducibility.
3.Empirical results confirm consistent improvements over prior datasets in multiple downstream benchmarks.

**Weaknesses:**

1.The novelty is primarily engineering-driven (data expansion) rather than conceptual.
2.The paper mainly focuses on CLIP-style retrieval; broader evaluations (e.g., text-to-molecule generation or reasoning tasks) are missing.
3.No systematic human evaluation or statistical verification of data correctness is reported.
4.Limited exploration of biases across molecular domains.

**Questions:**

1.How is factual correctness of generated molecular descriptions quantitatively validated?
2.Are there examples where GPT-4o-mini introduces incorrect or speculative statements, and how are they filtered?
3.What percentage of molecules lack valid synthesis or assay annotations after filtering?
4.Can you share the exact prompting strategy or scripts used for text generation?
5.How do generated texts compare qualitatively to real literature descriptions?
6.Have you analyzed inter-molecule text similarity to confirm linguistic diversity?
7.How would MolTextNet perform in cross-domain transfer?

---

### Official Review · Reviewer_dJnV · 2025-11-02

**Soundness:** 3
**Presentation:** 3
**Contribution:** 3
**Rating:** 2
**Confidence:** 2

**Summary:**

The authors present MolTextNet, a new, large-scale dataset of 2.5 million molecule-text pairs. The work is motivated by the significant limitations of existing datasets like PubChem-300K, which are relatively small-scale and, more importantly, contain extremely sparse and uninformative textual descriptions (e.g., a median length of 13 words). The core contribution is a synthetic data generation pipeline to create rich, informative descriptions. The authors start with 2.5 million molecules from ChEMBL35  and aggregate a comprehensive set of annotations for each molecule, including Structure, Properties, and Synthesizability.

This structured dictionary of annotations is then fed to an LLM (GPT-40-mini) using a specific prompt template. This prompt explicitly instructs the model to analyze and reason about the relationships between structure, properties, and synthesis, rather than simply listing the annotations. The resulting dataset has descriptions over 10 times longer on average than previous datasets.

**Strengths:**

- The paper addresses a clear and widely acknowledged bottleneck in multimodal molecular ML: the lack of large-scale, high-quality, and informative molecule-text data. A dataset of 2.5 million pairs with rich descriptions is a substantial contribution that could unlock new modeling capabilities.
- Good empirical results

**Weaknesses:**

- The most significant concern is that the entire text corpus is generated by an LLM. The paper trains models to align molecular graphs with a model's interpretation of chemical data, not with human-generated scientific text.
- The quality control section (3.3) is procedurally robust but semantically weak. The authors check for valid SMILES, deduplicate entries, and filter based on length or missing fields. However, there is no human expert validation of the generated text.
- The generator LLM (GPT-40-mini) was almost certainly trained on a corpus that includes PubChem, ChEMBL, and the OGB benchmarks. The authors state they filter explicit molecule overlaps with OGBG, which is good. However, this does not prevent conceptual leakage. The LLM's pre-existing knowledge of structure-property relationships (e.g., which scaffolds are common in HIV inhibitors) could be "leaked" into the generated text descriptions. This could, in turn, make the downstream OGBG-HIV task artificially easier, not because the GNN learned better chemical principles, but because the text-side provided "hints" that originate from the test set itself.
- The zero-shot retrieval task is a toy problem based on identifying simple functional groups. A much stronger validation would be to query for properties or bioactivity (e.g., "Retrieve molecules that inhibit DNA gyrase and have a low pKa"). This would actually test the novel property and synthesis information that MolTextNet claims to provide.

**Questions:**

See weaknesses

---

### Official Review · Reviewer_pqf7 · 2025-11-02

**Soundness:** 2
**Presentation:** 3
**Contribution:** 1
**Rating:** 2
**Confidence:** 4

**Summary:**

This paper presents MolTextNet, a dataset comprising over 2 million molecule-text pairs to support multimodal learning between molecular structures and language. The dataset is constructed by integrating captions generated from public databases (PubChem, ChEMBL) and augmenting missing textual descriptions using ChatGPT-assisted caption synthesis and RDKiT based properties. Further, the authors train foundation models on MolTextNet and evaluate performance on downstream tasks such as property prediction, text–molecule retrieval, and molecule captioning.

**Strengths:**

1. MolTextNet is impressive in size (~2 million molecule-caption pairs) and could serve as a valuable resource for pre-training large multimodal models in chemistry and biomedicine.
2. The data sourcing and pipeline design are clearly explained.
3. The design considerations are reasonable, by integrating information from a wide-range structural, chemical properties and integrating with functional annotations.

**Weaknesses:**

1. Reliability and hallucination concerns: A significant portion of captions are generated using ChatGPT, yet the paper provides no systematic validation or human evaluation of correctness. Prior work (e.g., MolTextQA) has reported high hallucination rates which raises serious concerns about factual accuracy. Larger synthetic datasets without validation may introduce noise rather than provide meaningful supervision
2. The dataset is not available for access, and the supplementary materials do not include even a small verification subset. For a dataset-driven paper, accessibility is critical to assess quality, and impact. Without at least a partial release, it is difficult to evaluate the contribution.
3. Reported performance gains over existing datasets such as PubChem-300K are small (often <1-2 %), and no statistical tests are presented to establish significance. This makes it unclear about the utility of the dataset.
4. The experiments focus on property prediction and molecule-text retrieval, but there is no evaluation on captioning quality itself, which is common with other works in the area.
5. The paper overlooks several directly relevant prior datasets:
    - MolQA (Xingyu et al., 2024) - a QA dataset with focus on factual validity.
    - MolTextQA (Laghuvarapu et al., 2025) – 500 K+ molecule-QA pairs connecting structure and textual semantics;
    - 3DMolLM (Sihang et al., 2024) – multimodal dataset incorporating molecular geometry and captions.
These works must be cited and compared in terms of dataset composition, along with any other missing related works.

[1] Lu, Xingyu, et al. "MoleculeQA: A dataset to evaluate factual accuracy in molecular comprehension." arXiv preprint arXiv:2403.08192 (2024).
[2] Laghuvarapu, Siddhartha, et al. "MolTextQA: A Question-Answering Dataset and Benchmark for Evaluating Multimodal Architectures and LLMs on Molecular Structure–Text Understanding." Journal of Data-centric Machine Learning Research (2025).
[3] Li, Sihang, et al. "Towards 3d molecule-text interpretation in language models." arXiv preprint arXiv:2401.13923 (2024).

**Questions:**

1. Have you performed any human evaluation of the generated captions’ correctness?
2. What procedures were used to detect or reduce hallucinations in ChatGPT-generated text? Were chemical rules, filters, or expert checks applied?
3. Are the reported improvements over PubChem-300K statistically significant? Could you provide significance tests?
4. What are the licensing and hosting plans for the dataset ?

---

### Note · Authors · 2025-12-03

I have read and agree with the venue's withdrawal policy on behalf of myself and my co-authors.